# Filament formation by metabolic enzymes is a specific adaptation to an advanced state of cellular starvation

Ivana Petrovska[1†], Elisabeth Nüske[1†], Matthias C Munder[1], Gayathrie Kulasegaran[1], Liliana Malinovska[1], Sonja Kroschwald[1], Doris Richter[1], Karim Fahmy[2], Kimberley Gibson[1], Jean-Marc Verbavatz[1], Simon Alberti[1*]

[1]Max Planck Institute of Molecular Cell Biology and Genetics, Dresden, Germany; [2]Institute of Resource Ecology, Helmholtz Institute Dresden-Rossendorf, Dresden, Germany

**Abstract** One of the key questions in biology is how the metabolism of a cell responds to changes in the environment. In budding yeast, starvation causes a drop in intracellular pH, but the functional role of this pH change is not well understood. Here, we show that the enzyme glutamine synthetase (Gln1) forms filaments at low pH and that filament formation leads to enzymatic inactivation. Filament formation by Gln1 is a highly cooperative process, strongly dependent on macromolecular crowding, and involves back-to-back stacking of cylindrical homo-decamers into filaments that associate laterally to form higher order fibrils. Other metabolic enzymes also assemble into filaments at low pH. Hence, we propose that filament formation is a general mechanism to inactivate and store key metabolic enzymes during a state of advanced cellular starvation. These findings have broad implications for understanding the interplay between nutritional stress, the metabolism and the physical organization of a cell.

*For correspondence: alberti@mpi-cbg.de

†These authors contributed equally to this work

Competing interests: The authors declare that no competing interests exist.

## Introduction

The organization of the cytoplasm into functionally distinct compartments is fundamental to life. Eukaryotes in particular have achieved a high level of organizational complexity by restricting specific biochemical reactions to membrane-bound organelles in the cytoplasm. A different but equally effective strategy for localized biochemistry involves the formation of phase-separated macromolecular assemblies, which are built de novo from proteins and RNAs (*Brangwynne, 2011*; *Hyman and Simons, 2012*; *O'Connell et al., 2012*; *Wilson and Gitai, 2013*). These non membrane-bound assemblies are often large, occupying an intermediate length scale situated between the nanoscale of individual macromolecules and the microscale of cells. Such intermediate-sized (or mesoscale) assemblies often have specific functions that are intricately linked to their higher order complexity.

Mesoscale assemblies are structurally and functionally diverse, ranging from dynamic RNA granules, such a P bodies and stress granules, to highly ordered microcompartments, such as bacterial carboxysomes. Based on their physicochemical properties they can be categorized into two distinct groups: liquid-like and crystalline. Liquid-like assemblies are disordered and very dynamic (the components turn over on the order of seconds to minutes) (*Brangwynne et al., 2009*, *2011*; *Li et al., 2012*), whereas crystalline assemblies are more ordered and static (*Yeates et al., 2010*; *Wilson and Gitai, 2013*). The distinctive properties of these assemblies also entail different geometries: liquid-like assemblies are usually spherical, whereas crystalline assemblies adopt a broader variety of shapes, ranging from linear polymers and two-dimensional polymer sheets to extensively cross-linked three-dimensional crystals. Modern imaging techniques are now enabling us to unravel the

**eLife digest** Life is based on a series of chemical reactions that control how cells live, grow, and divide. Various metabolic enzymes allow cells to control the rate at which these reactions occur. Recently, researchers have noticed that metabolic enzymes can form filaments in cells, usually when the cells are deprived of energy or nutrients. Petrovska, Nüske et al. now reveal more about how and why an enzyme called glutamine synthetase (Gln1) forms filaments in yeast cells.

Gln1 has a cylindrical shape. This shape means that stacking the enzymes end-to-end under the right conditions is enough to make them bond into a long filament. In addition, a zip-like mechanism enables neighboring filaments to fuse to create thicker fibres. These filaments are more likely to form if there are high concentrations of large background molecules around—a condition known as macromolecular crowding.

Petrovska, Nüske et al. also found evidence that the filaments are part of a strategy to help cells survive starvation. Enzymes were more likely to construct filaments when the cell division cycle had stopped, which commonly occurs due to a lack of nutrients. In addition, Gln1 filaments only form if the cytoplasm of the cell becomes acidic—which is a response to the cell starving. This has been seen for other metabolic enzymes as well, suggesting that acidification is a signal to reprogram the metabolism of a cell.

The Gln1 enzymes in a filament are inactivated, but become active again after the filament breaks up. In addition, preventing Gln1 filament formation makes it harder for cells to recover from a period of starvation. Petrovska, Nüske et al. therefore suggest that the filaments act as a storage depot for the enzymes during starvation. Further experiments are now needed to uncover exactly how these manage to help the starving cell to survive and recover.

various ways in which nature uses such organizational principles to functionally diversify the sub-cellular landscape.

The number of identified mesoscale assemblies in prokaryotic and eukaryotic cells is increasing rapidly (*Yeates et al., 2010*; *Hyman and Simons, 2012*; *Wilson and Gitai, 2013*). Metabolic enzymes in particular are able to self-assemble into higher order structures (*O'Connell et al., 2012*), suggesting an important role in regulating the metabolism of cells. These findings establish a link to earlier work from Paul Srere and colleagues, who proposed that metabolic enzymes undergo static and dynamic interactions to spatiotemporally organize and regulate metabolic pathways (*Srere, 1987*). Such organizational specificity was believed to be necessary to ensure metabolic efficiency. Additional studies suggest that self-assembling metabolic enzymes can be co-opted for other functional roles. A particularly striking example is that of cytidine triphosphate synthase, a bacterial enzyme that forms cytoskeletal filaments with cell morphological functions (*Ingerson-Mahar et al., 2010*; *Barry and Gitai, 2011*).

Recent large-scale studies in budding yeast identified several metabolic enzymes that form punctate or filamentous structures in the cytoplasm. Examples include cytidine triphosphate synthase (*Noree et al., 2010*), an enzyme involved in the synthesis of cytosine nucleotides, and glutamine synthetase (*Narayanaswamy et al., 2009*), an enzyme that promotes the conversion of glutamate into glutamine. The metabolic reactions catalyzed by these newly identified enzymes were diverse, but the conditions under which they assembled appeared to be similar: filament formation was most frequently observed, when cells entered stationary phase or were depleted of their primary energy source: glucose. This suggests a functional association with the energy-depleted cellular state, an assumption that has so far remained untested. Moreover, it is not yet clear whether these assemblies are catalytically active, serve as enzyme storage compartments, or are protein aggregates.

We conducted a comprehensive analysis of the molecular underpinnings and functions of filaments formed by metabolic enzymes during advanced conditions of starvation. Using yeast glutamine synthetase (Gln1) as a model enzyme, we show that filament formation involves the repeated stacking of homo-decameric enzyme complexes by a back-to-back mechanism. We further demonstrate that filament assembly is triggered by a starvation-induced drop in the intracellular pH and results in enzymatic inactivation and the formation of enzyme storage depots. Importantly, these findings also extend to other filament-forming proteins, arguing that filament formation by metabolic enzymes is a specific adaptation that allows yeast to endure and recover from severe starvation conditions.

## Results

### Gln1 assembles into filaments in starved yeast cells

Gln1 is an essential metabolic enzyme that catalyzes the ATP-dependent synthesis of glutamine from glutamate and ammonium. In agreement with previous studies (*Narayanaswamy et al., 2009*), we found that GFP-tagged Gln1 was evenly distributed in dividing cells but coalesced into fluorescent foci in cells that were maintained in a phosphate buffer (*Figure 1—figure supplement 1*). While most of these foci had a punctate appearance, we noticed a few that displayed a rod-like shape. This suggested that Gln1 might be able to assemble into filamentous structures, but at the same time it implied that the efficiency of filament formation was very low.

To exclude a potential interference from the GFP tag, we made use of the fact that the homo-oligomeric structure of Gln1 allows mixed oligomers composed of tagged and untagged monomers. Indeed, when we co-expressed untagged and GFP-tagged Gln1 in yeast, the punctate localization pattern disappeared, and Gln1-GFP assembled into a few filaments per cell (*Figure 1—figure supplement 2*). To verify this finding, we replaced endogenous Gln1 with a version containing a C-terminal tetracystein tag. This stretch of six amino acids can be labeled with small dyes such as FlAsH (*Adams et al., 2002*). Indeed, FlAsH-labeled Gln1 assembled into filaments in cells that were transferred from media to buffer (*Figure 1—figure supplement 3*). In a search for a protein-encoded fluorescent tag that is compatible with the filamentous state, we replaced the GFP tag in the endogenous *GLN1* locus with mCherry. Indeed, unlike GFP-tagged Gln1, mCherry-tagged Gln1 assembled into filaments (*Figure 1A*). The number of filaments per cell as well as the kinetics of filament formation was comparable to our previous experiment with predominantly untagged Gln1 (*Figure 1—figure supplement 2*). These data indicate that mCherry is compatible with the filamentous state and therefore a suitable fluorophore to study the localization of Gln1 in living cells.

Using live cell microscopy, we found that mCherry-tagged Gln1 was diffusely localized in dividing cells but formed filaments when the growth medium lacked a carbon source (33% of the cells had filaments after 4 hr of glucose starvation) (*Figure 1B*). Importantly, when we transferred the cells into a phosphate buffer that contained no metabolizable nutrients, filaments were detectable in all cells (*Figure 1C*). Thus, under conditions of severe starvation, the filament formation phenotype becomes fully penetrant. This suggests that filament formation by metabolic enzymes is a starvation-induced cellular adaptation. Here, we refer to this specific cellular state as the state of advanced starvation.

On average, filament assembly started 50 min (n = 179; SD = 43.9 min) after onset of advanced starvation conditions (*Figure 1D* and *Video 1*). However, we observed extensive variation from cell to cell, suggesting that yeast vary in their ability to deal with sudden energy depletion (*Video 2*). Importantly, addition of glucose to buffer was sufficient to prevent filament formation (*Figure 1C*). We also found that cells can fully recover from the state of advanced starvation. Upon resupply of nutrients, filament dissolved very rapidly, and shortly after filament dissolution cells re-entered into the cell cycle (*Figure 1E*, *Videos 3 and 4*). On average, filament dissolution was completed ~18 min after nutrient addition (n = 65; SD = 11.6 min). We conclude that Gln1 dynamically and reversibly assembles into filaments when yeast cells are exposed to conditions of advanced starvation.

### Specific point mutations in Gln1 inhibit or promote filament formation

A recent study reported a high-resolution crystal structure for yeast Gln1 (*He et al., 2009*). The enzyme is composed of two pentameric rings that interact to form a face-to-face homo-decamer. This structural arrangement is identical to those of mammalian and plant glutamine synthetases. Curiously, however, the crystal structure of yeast Gln1 also revealed a new back-to-back association between two decamers (*Figure 2A*). The biological significance of this novel interface, however, remained undetermined.

The presence of the decamer–decamer interface led us to hypothesize a simple mechanism for filament formation: assembly by repeated back-to-back stacking of decamers. To test this assumption, we mutated specific residues in the interface (*Figure 2A* and *Figure 2—figure supplements 1, 2 and 3*). The resulting Gln1 variants were fused to mCherry and tested in yeast for their ability to assemble into filaments. We found three mutations (E186K, P83R, T49E) that abrogated the ability of Gln1 to form filaments in starved cells (*Figure 2B*). Intriguingly, we also identified two mutants that formed filaments in growing cells. The Y81A mutant had a slightly increased propensity to form filaments, whereas the R23E mutant formed filaments constitutively, regardless of the growth conditions. To test whether the R23E mutant effect could be neutralized by a second mutation, we introduced the inhibitory T49E

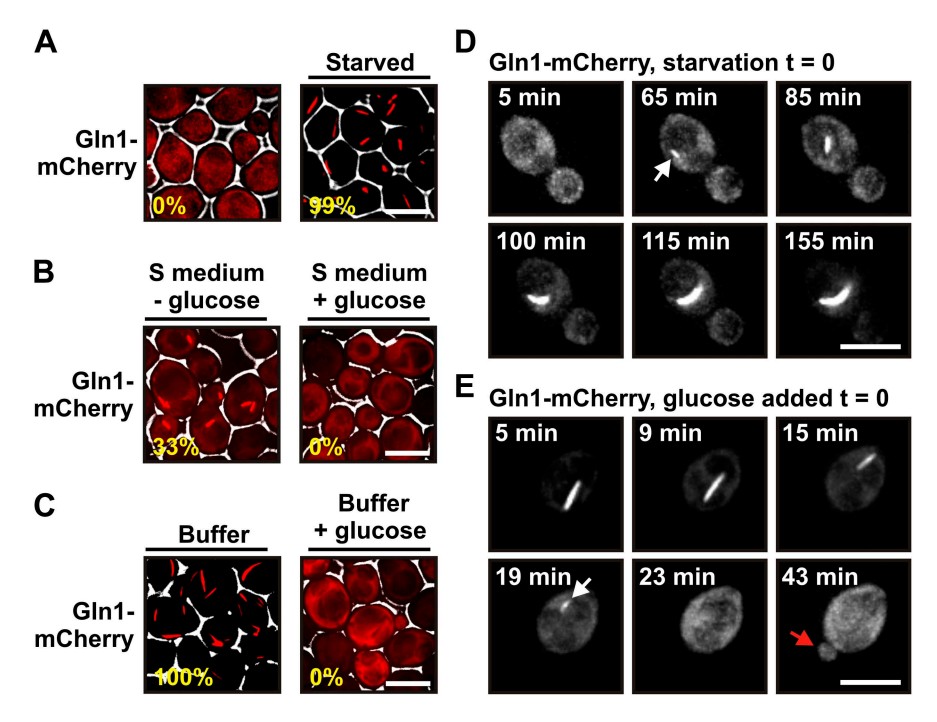

**Figure 1**. Gln1 assembles into filaments in energy-depleted yeast cells. (**A**) Yeast cells expressing mCherry-tagged Gln1 from the endogenous promoter were washed twice with water and resuspended in synthetic media (left, control) or citrate buffer of pH 6 (right, 'starved'). White lines are the cell boundaries. The scale bar is 5 μm. The numbers in yellow give the percentage of cells with fluorescent foci. At least 200 cells were counted. (**B**) Log phase yeast cells expressing mCherry-tagged Gln1 were washed twice with water and resuspended in synthetic media without (left) or with (right) 2% glucose. Images were taken 4 hr after onset of glucose starvation. (**C**) Log phase cells expressing mCherry-tagged Gln1 were washed twice with water and resuspended in a phosphate–citrate buffer of pH 6 without (left) or with (right) 2% glucose. Images were taken 4 hr after onset of starvation. (**D**) Cells expressing Gln1-mCherry were washed twice with water and resuspended in a phosphate–citrate buffer of pH 6 to induce starvation (time point 0). Filament formation was followed by time-lapse microscopy. Individual time points are indicated in minutes. The white arrow designates an emerging filament. The scale bar is 5 μm. Also see the corresponding *Video 1*. (**E**) Same as (**D**) except that filament dissolution was investigated by re-adding glucose to cells that had been starved for 4 hr. The white arrow points to a small filament. The red arrow designates the emerging bud. Also see the corresponding *Video 3*.

The following figure supplements are available for figure 1:

**Figure supplement 1**. GFP-tagged Gln1 predominantly forms punctate structures.

**Figure supplement 2**. Co-expression of untagged Gln1 transforms the localization pattern from punctate to filamentous.

**Figure supplement 3**. Filamentation is not caused by the tag.

mutation into the R23E variant. Indeed, the double mutant (R23E, T49E) was no longer able to form filaments (*Figure 2B*), indicating that the increased propensity to form filaments can be overcome by an inhibitory second site mutation. These findings indicate that the propensity to form filaments can be modulated by specific mutations in Gln1. They also suggest that the decamer–decamer interface is biologically important.

## Gln1 assembles by a back-to-back stacking mechanism

To provide further evidence for the proposed mechanism of assembly, we purified wild-type and mutant Gln1 from yeast cells. From this point on we will use a color code to facilitate discrimination

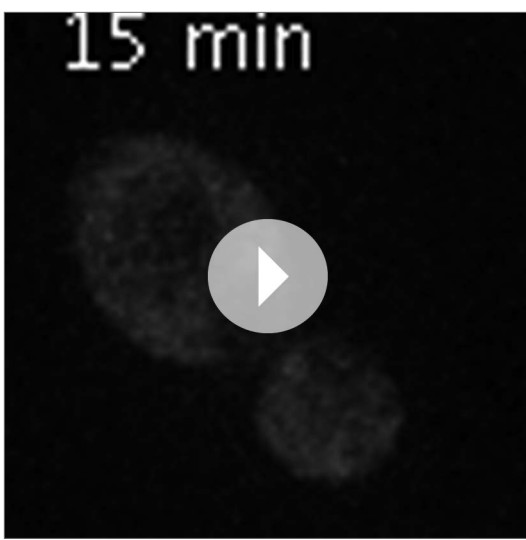

**Video 1**. Gln1 forms filaments in starved yeast. Cells expressing Gln1-mCherry were washed twice with water and resuspended in a phosphate-citrate buffer of pH 6 to induce starvation (time point 0). Filament formation was followed by time-lapse microscopy. Time points are indicated in minutes.

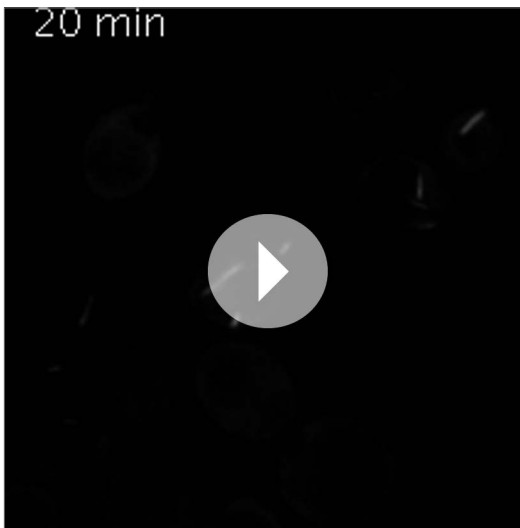

**Video 2**. Gln1 forms filaments in starved yeast. Cells expressing Gln1-mCherry were washed twice with water and resuspended in a phosphate-citrate buffer of pH 6 to induce starvation (time point 0). Filament formation was followed by time-lapse microscopy. Time points are indicated in minutes.

between the different mutant forms of Gln1. Mutants with inhibitory effects on filament formation will be highlighted in red, whereas constitutively filament-forming mutants will be highlighted in green. First, we investigated the oligomeric state of wild-type and variant Gln1 by blue native PAGE and immunoblotting. As can be seen in *Figure 2C*, the predominant form of Gln1 was a decamer. Importantly, however, we also identified distinct bands that corresponded to higher molecular weight forms of the enzyme. These assemblies were predominantly detectable for the wild-type and the R23E variant. Moreover, when compared to wild-type Gln1, the constitutively filament-forming R23E variant showed an increased number of these higher order assemblies.

To eliminate the possibility that formation of these assemblies involves cross-β structure—a hallmark of other filament-forming proteins such as Pmel17 (*Fowler et al., 2006*) or Curli (*Chapman et al., 2002*)—we performed two additional control experiments. First, we determined whether Gln1 filaments could be stained with Thioflavin T (a dye that specifically binds to cross-β structure), and second, we subjected lysates from filament-containing cells to semi-denaturing detergent–agarose gel electrophoresis (a method that detects cross-β structure based on its resistance to detergent) (*Alberti et al., 2010*). As can be seen in *Figure 2—figure supplements 4 and 5*, we could not find evidence for cross-β structure. As a next step, we isolated His-tagged R23E Gln1 from yeast cells and investigated the purified protein by negative staining and electron microscopy. We identified short filamentous structures, which, upon closer inspection, revealed that they were formed by a repeating unit that precisely matched the dimensions of a Gln1 decamer (*Figure 2—figure supplement 6*). Thus, we conclude that Gln1 retains a near-native structure when it assembles into filaments.

As a next step, we performed correlative light and electron microscopy (CLEM) experiments on cells overexpressing the R23E variant as mCherry fusion. The ultrastructural features of the mCherry-positive structures are shown in *Figure 2D*. The electron micrographs show a large number of filaments, which are laterally aligned into higher order bundles. This side-by-side bundling is consistent with the growth pattern of the filaments, which was predominately in the longitudinal direction but also included some increase in circumference over time

(see *Video 1*). To exclude that the bundling was caused by the mCherry tag, we performed a careful ultrastructural analysis of yeast cells that expressed untagged R23E. Indeed, we could also identify fibrillar structures in the cytoplasm of R23E-expressing cells but not in control cells (*Figure 2—figure supplement 7*). However, we noticed that the filaments were in closer contact, suggesting that the mCherry tag has

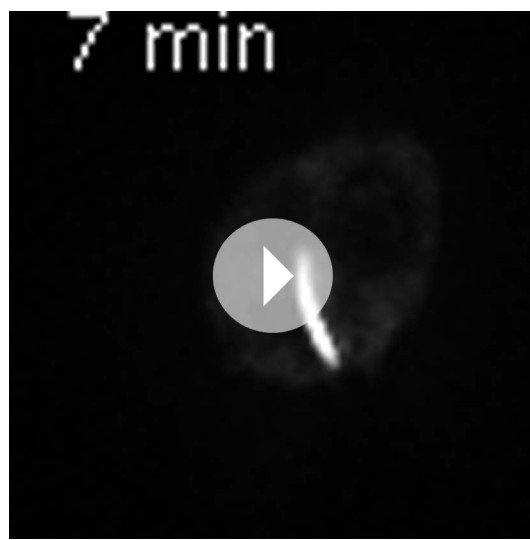

**Video 3**. Gln1 filaments dissolve upon glucose addition to starved cells. Cells expressing Gln1-mCherry were washed twice with water and resuspended in a phosphate-citrate buffer of pH 6 to induce starvation (time point 0). The cells were incubated for 4 hr to induce the formation of filaments. At time point 0, glucose (2%) was added and filament dissolution was followed by time-lapse microscopy.

**Video 4**. Gln1 filaments dissolve upon glucose addition to starved cells. Cells expressing Gln1-mCherry were washed twice with water and resuspended in a phosphate-citrate buffer of pH 6 to induce starvation (time point 0). The cells were incubated for 4 hr to induce the formation of filaments. At time point 0, glucose (2%) was added and filament dissolution was followed by time-lapse microscopy.

some influence on the packing density. Together, these findings indicate that Gln1 assembles into filaments by a back-to-back stacking mechanism. Once formed, these filaments can organize into higher order fibrils.

## Self-assembly into filaments is driven by macromolecular crowding

Is Gln1 able to self-assemble or does it need additional factors? To investigate this question, we purified wild-type and variant Gln1 from bacteria. The purified proteins were investigated by dynamic light scattering and gel exclusion chromatography for their ability to assemble into high molecular weight forms. As can be seen in *Figure 3A* and *Figure 3—figure supplement 1*, R23E Gln1 had a strongly increased propensity to assemble into higher order forms. This propensity was reduced in wild-type Gln1 and absent from a variant that had lost the ability to assemble into filaments in yeast cells (E186K). We next analyzed purified R23E Gln1 by electron microscopy (*Figure 3—figure supplement 2*). Our analysis revealed abundant cylindrical particles, consistent with the reported structure of Gln1 (*He et al., 2009*). Interestingly, these particles were organized into chains, providing further support for the proposed back-to-back assembly mechanism. However, we were unable to identify higher order structures that resembled previously observed fibrillar structures in yeast, suggesting that an important factor was missing.

What could be the missing factor? A first hint came from our attempts to purify filaments from yeast cells. When we lysed filament-containing yeast, the filaments were unstable (*Figure 3B*). To identify components that are necessary for filament integrity, we performed experiments with modified lysis buffers. One obvious difference to yeast cytoplasm was that the lysis buffer lacked a high background concentration of macromolecules. We therefore tested the influence of macromolecular crowders on filament stability. Strikingly, when we added the crowding agent Ficoll 70 at a concentration close to the physiological concentration of macromolecules (200 mg/ml), the filaments remained intact (*Figure 3C*).

To further investigate the role of macromolecular crowding, we purified mCherry-tagged R23E from yeast cells and incubated it for 1 hr in the presence or absence of a crowder. The samples were subsequently analyzed by fluorescence microscopy for the formation of higher order structures. Intriguingly, we found that R23E Gln1 formed filamentous structures in the presence of a crowder but not in its absence (*Figure 3D*). Importantly, similar structures could not be detected when we replaced R23E with the assembly-incompetent

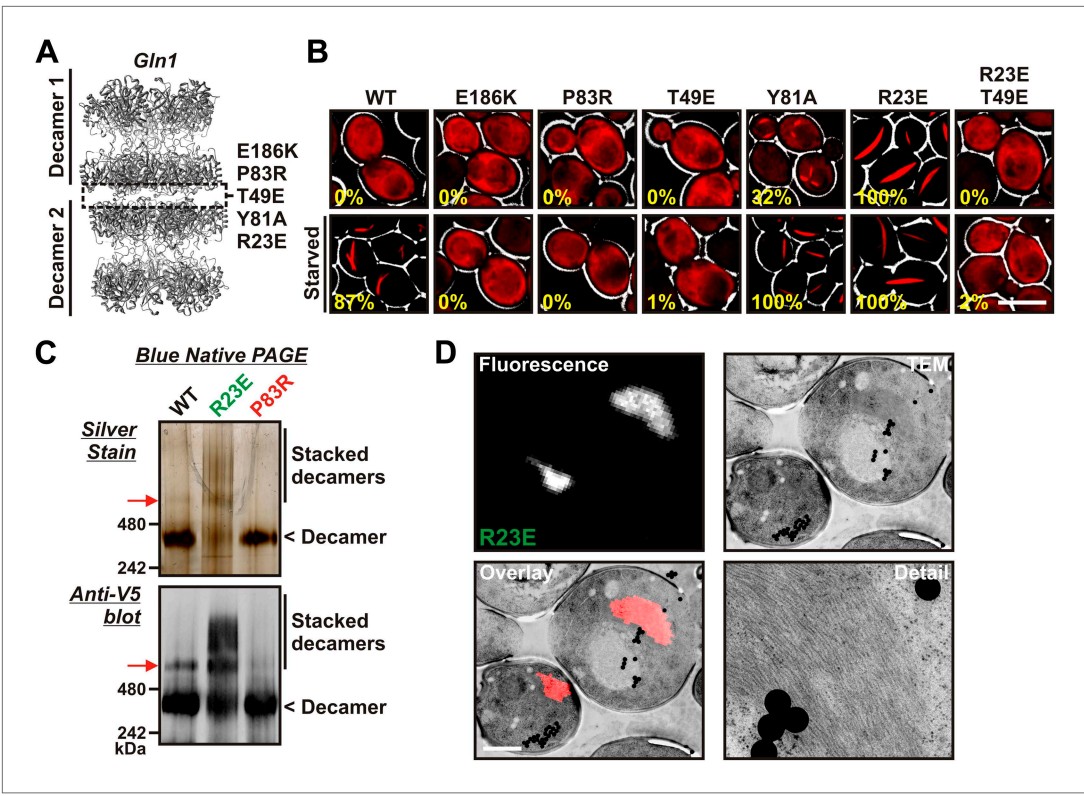

**Figure 2**. Gln1 assembles by a back-to-back stacking mechanism. (**A**) The crystal structure of Gln1 as reported by *He at al. (2009)*. The putative assembly interface and mutations introduced in this study are indicated. (**B**) Chromosomally encoded Gln1 was replaced with mCherry-tagged wild-type or variant Gln1 expressed from a plasmid. The strains were washed twice with water and resuspended in synthetic media (top, control) or buffer (bottom, 'starved'). The numbers in yellow give the percentage of cells with fluorescent foci. At least 200 cells were counted. The scale bar is 5 μm. (**C**) Wild type or variant 6xHis-Gln1-V5 was purified from yeast and analyzed by blue native PAGE. Proteins were detected by silver staining (top) or immunoblotting with an antibody that recognized a C-terminal V5 tag (bottom). The red arrow denotes a band that corresponds to two stacked decamers. The calculated size of a Gln1 decamer is 420 KDa. (**D**) Correlative light electron microscopy (CLEM) was performed on yeast cells expressing the R23E variant of Gln1 as mCherry fusion. The black dots are fluorescent beads, which were introduced to facilitate the alignment of the fluorescence and TEM images. The scale bar is 500 μm.

The following figure supplements are available for figure 2:

**Figure supplement 1**. Detailed structural view of the decamer–decamer interface.

**Figure supplement 2**. Detailed structural view of the decamer–decamer interface.

**Figure supplement 3**. Detailed structural view of the decamer–decamer interface.

**Figure supplement 4**. Yeast cells expressing mCherry-tagged R23E Gln1 were subjected to staining with Thioflavin T.

**Figure supplement 5**. Lysates from yeast cells expressing wild-type or R23E Gln1 were subjected to semi-denaturing detergent-agarose gel electrophoresis (SDD-AGE).

**Figure supplement 6**. 6xHis-Gln1(R23E)-mCherry was affinity purified from yeast, subjected to negative staining and investigated by electron microscopy.

**Figure supplement 7**. Transmission electron microscopy (TEM) was performed on yeast cells expressing untagged Gln1(R23E).

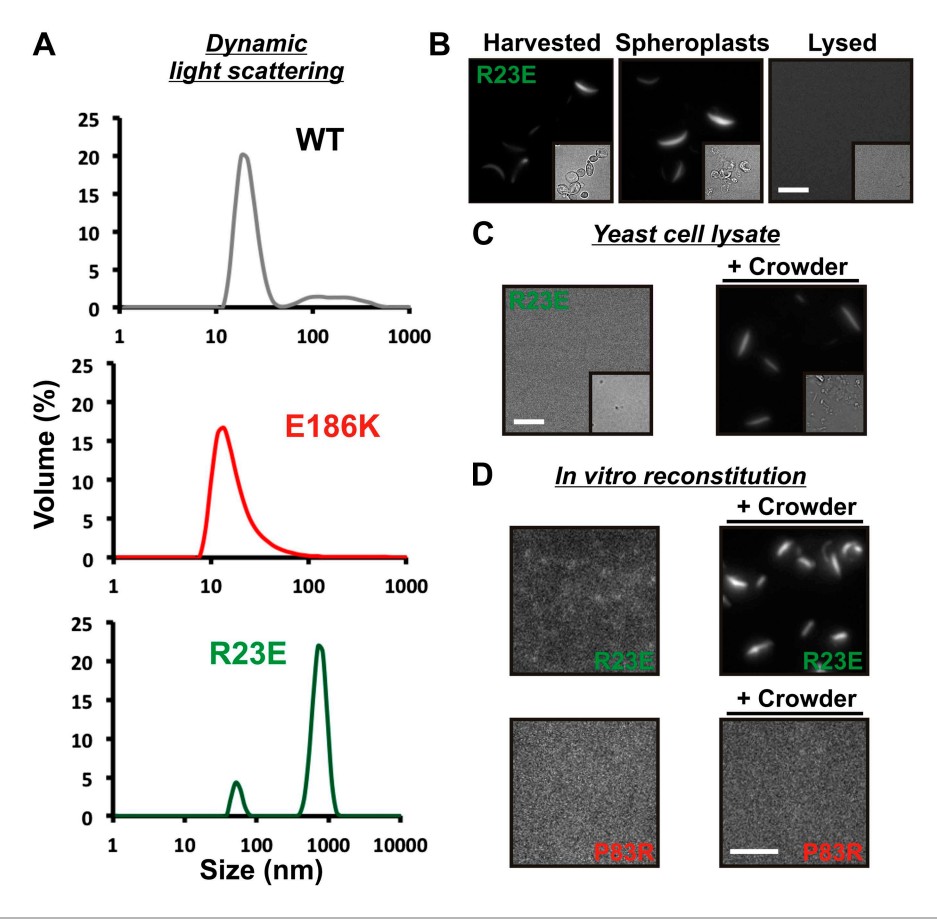

**Figure 3**. Self-assembly into filaments is driven by macromolecular crowding. (**A**) Equal amounts of 6xHis-tagged wild-type and variant Gln1 purified from bacteria were subjected to dynamic light scattering. Shown is the volume distribution that was derived from the intensity distribution. Note the different scales of the x axes. (**B**) Yeast cells expressing Gln1(R23E)-mCherry were spheroplasted and lysed. Images were acquired from harvested, spheroplasted, and lysed cells. Images were taken at the same intensity settings. The inset is the corresponding DIC image. The scale bar is 5 μm. (**C**) Cells were treated as in (**A**) except that the lysis buffer contained Ficoll 70 at a concentration of 200 mg/ml. (**D**) Gln1-mCherry was purified from yeast and incubated in a phosphate-citrate buffer of pH 7 with or without a crowding agent for 1 hr. Samples were analyzed by fluorescence microscopy and images were taken at the same intensity settings. The scale bar is 3 μm.

The following figure supplements are available for figure 3:

**Figure supplement 1**. Gel filtration of wild type and variant 6xHis-tagged Gln1 purified from bacteria.

**Figure supplement 2**. Electron microscopy of 6xHis-tagged R23E Gln1 purified from bacteria.

**Figure supplement 3**. 6xHis-Gln1-mCherry was affinity purified from yeast and assembled in the presence of a crowder.

variant P83R. Further inspection of in vitro formed Gln1 filaments revealed structures of varying thickness (**Figure 3—figure supplement 3**). This suggests that in vitro reconstituted filaments are able to assemble into higher order bundles, as in yeast cells. Thus, we conclude that the assembly of Gln1 is strongly dependent on macromolecular crowding but independent of other cellular components.

## A drop in intracellular pH triggers filament formation

Wild-type Gln1 only formed filaments in energy-depleted cells, in contrast to the R23E variant, which assembled also in dividing cells. This raised questions about the trigger for assembly in starved yeast.

Again experiments with lysates of R23E-expressing cells were revealing. These experiments showed that the stability of R23E filaments could not only be increased by crowders but also by acidifying the lysis buffer. A cell lysate prepared with a lysis buffer of pH 5 contained abundant filaments, while the filaments began to fall apart when the lysis buffer was adjusted to pH 6, and they were absent from lysates adjusted to pH 7 or 8 (*Figure 4A*). This raised the possibility that intracellular pH changes are the trigger for filament formation.

The intracellular pH of yeast cells drops significantly when yeast are depleted of their primary energy source glucose (*Orij et al., 2009*; *Dechant et al., 2010*; *Orij et al., 2011*). This is because yeast cells have to continuously expend energy to maintain the proton gradient across the plasma membrane (usually yeast media are acidic). To test whether pH changes are sufficient to trigger the assembly of Gln1, we artificially acidified the cytoplasm by adding the protonophore 2,4-dinitrophenol (DNP) to dividing yeast. Strikingly, acidified yeast abundantly formed filamentous structures, despite the presence of glucose (*Figure 4B*). Next, we tested whether a neutral or basic outside pH could prevent the formation of filaments in energy-depleted yeast. Indeed, starved yeast maintained in a buffer of pH 7 contained shorter and smaller filaments than yeast in a

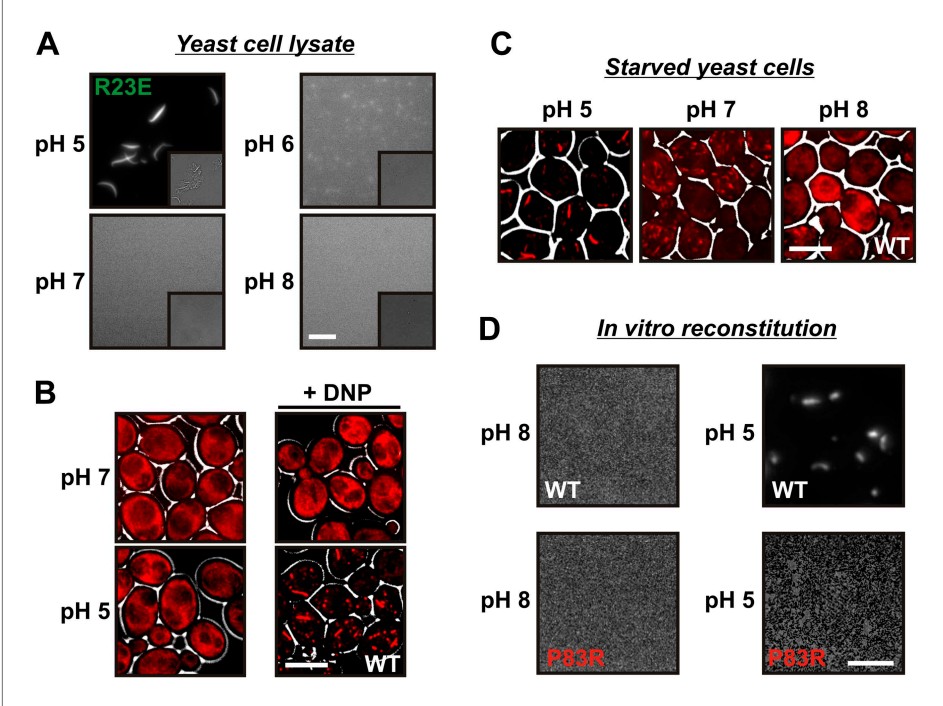

**Figure 4**. A drop in intracellular pH triggers filament formation. (**A**) Yeast cells expressing Gln1(R23E)-mCherry were spheroplasted and lysed in phosphate buffers of different pHs. Images were acquired immediately after lysis. The scale bar is 5 μm. Note that the lysis buffer did not contain a crowder. (**B**) Yeast cells expressing mCherry-tagged Gln1 were washed twice with water and resuspended in a glucose-containing buffer of the indicated pHs with or without the proton carrier 2,4-dinitrophenol (DNP). Images were taken 1 hr after addition of the buffer. The scale bar is 5 μm. (**C**) Yeast cells expressing Gln1-mCherry were washed twice with water and resuspended in phosphate buffers of different pHs to induce starvation. The buffers contained proton carriers and energy inhibitors for rapid equilibration of inside and outside pH. The scale bar is 5 μm. (**D**) Gln1-mCherry was purified from yeast and incubated in an acidic or basic buffer containing a crowding agent for 1 hr. Samples were analyzed by fluorescence microscopy. The scale bar is 3 μm.

The following figure supplements are available for figure 4:

**Figure supplement 1**. Equal amounts of 6xHis-tagged wild type and E186K Gln1 purified from bacteria were mixed with an acidic buffer and subjected to dynamic light scattering.

**Figure supplement 2**. 6xHis-tagged wild-type Gln1 purified from bacteria was subjected to Far-UV CD at pH 7.4 and 6.

buffer of pH 5 (*Figure 4C*). Moreover, filaments were largely absent when the buffer was adjusted to pH 8.

Intracellular pH changes could directly or indirectly affect the assembly of Gln1. To differentiate between these two possibilities, we purified mCherry-tagged Gln1 from yeast and incubated it for 1 hr in a buffer of pH 5 or 8. Afterwards, the samples were examined by fluorescence microscopy for the formation of higher order structures. Indeed, Gln1 formed filamentous structures in the acidic buffer but not in the control buffer with a basic pH (*Figure 4D*). Importantly, this effect was specific, because the assembly-deficient variant P83R did not assemble into higher order structures in an acidic buffer. To follow pH-induced assembly over time, we exposed bacterially purified Gln1 to an acidic buffer and monitored the formation of higher order structures by dynamic light scattering. As can be seen in *Figure 4—figure supplement 1*, Gln1 progressively assembled into higher order structures, whereas a mutant version of Gln1 (E186K) did not. Moreover, exposure to an acidic buffer did not lead to an extensive alteration of the secondary structure of Gln1 (*Figure 4—figure supplement 2*), suggesting that the protein retains its structural integrity under these conditions. Thus, we conclude that Gln1 assembly is directly regulated by protons and that the trigger for filament formation in yeast cells is a starvation-induced drop in the intracellular pH.

## Acidification of the cytosol is a general trigger for filament formation

A recent study reported that several protein complexes assemble into filaments as yeast cells approach stationary phase (*Noree et al., 2010*). One of these filaments consisted of only one protein (Glt1), while the others were composed of multiple proteins with two (Ura7/Ura8) or more (Gcd2, Gcd6, Gcd7, Gcn3, Sui2) distinct subunits. To study whether cytosolic acidification is also required for formation of these assemblies, we tagged one protein in each filament with GFP. We then determined the frequency of filaments in yeast cells that were resuspended in different pH-adjusted buffers. Because these buffers lacked an energy source, the starved cells rapidly adopted the outside pH. Intriguingly, we found that all three proteins formed filaments in a strongly pH-dependent manner (*Figure 5A*). Moreover, filament formation was almost completely abrogated at a neutral or basic pH. We then tested if we could induce filament formation in growing cells by adding the protonophore DNP. Indeed, cells suspended in a DNP-containing buffer of pH 5 or pH 6 showed widespread filament formation, despite the presence of glucose (*Figure 5B*). However, filaments were absent from cells maintained in a DNP-containing buffer of pH 7. This indicates that pH changes are a general trigger for the formation of filaments by metabolic enzymes.

## Assembled Gln1 is inactive but becomes active again after disassembly

To investigate the functional implications of filament formation, we first tested whether filament formation modulates the activity of Gln1. Indeed, Gln1 was progressively inactivated under conditions of advanced starvation (*Figure 6A*), and the extent of inactivation correlated with the amount of filament formation (*Figure 6—figure supplement 1*). Next, we compared the enzymatic activities of wild-type and mutant Gln1. As can be seen in *Figure 6B*, the R23E variant displayed a strongly reduced synthetase activity. Because these activity measurements were performed with cell lysates, we hypothesized that the actual enzymatic inactivation is much stronger in intact cells. To investigate this possibility, we inspected wild-type and variant Gln1 variants for growth phenotypes. Consistent with the essential function of Gln1, the constitutively assembling R23E variant had a strongly diminished ability to grow (*Figure 6C*). Importantly, this growth deficiency was abrogated by a second site mutation that prevents filament formation (T49E), indicating that filament formation is the cause of the reduced growth rate. Moreover, cells growing on glutamine as the sole nitrogen source (*Figure 6—figure supplement 2*) showed no growth defect, suggesting that this deficiency was due to the inability of the cells to produce sufficient amounts of the growth-promoting amino acid glutamine. Based on these findings, we conclude that filament formation during advanced starvation conditions silences the enzymatic activity of Gln1.

Do these inactive filaments serve as storage depots for Gln1? To address this question, we generated a yeast strain that expressed mCherry-tagged Gln1 from the inducible *GAL* promoter. Because of the essential nature of Gln1, this strain had to be maintained in the presence of galactose. To test whether assembled Gln1 can become enzymatically active again, we grew the strain in the presence of galactose and then induced starvation by transferring it into a phosphate buffer. After 4 hr of starvation the great majority of the cells had formed filaments. We then added glucose-containing growth

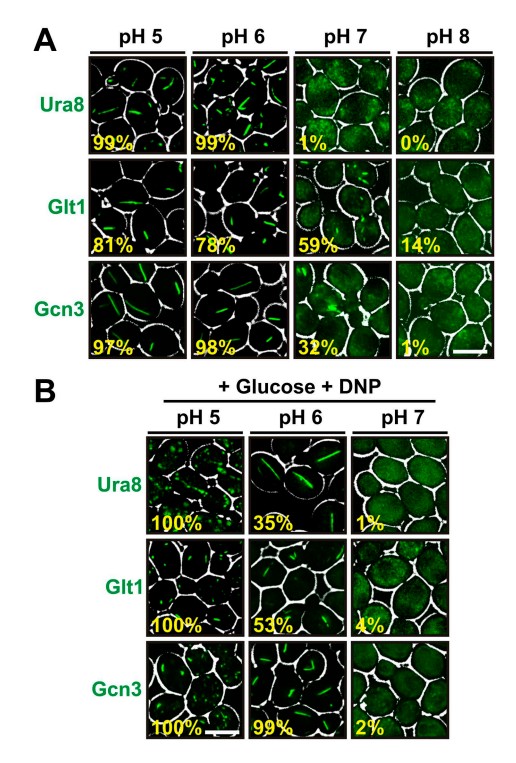

**Figure 5**. Other metabolic enzymes form filaments in a pH-dependent manner. (**A**) Yeast cells expressing sfGFP(V206R)-tagged Ura8, Glt1, or Gcn3 were washed twice with water and resuspended in buffers of different pHs to induce starvation. Images were taken 2 hr after onset of starvation. The scale bar is 5 μm. (**B**) Yeast cells expressing sfGFP(V206R)-tagged Ura8, Glt1, or Gcn3 were washed twice with water and resuspended in a buffer containing 2% glucose and the proton carrier 2,4-dinitrophenol (DNP). Images were taken 1 hr after addition of the buffer. The numbers in yellow give the percentage of cells with fluorescent foci. At least 200 cells were counted. The scale bar is 5 μm.

medium to induce reentry into the cell cycle. Importantly, under these conditions, the *GAL* promoter is switched off, and Gln1 is no longer synthesized de novo. Therefore, the cells can only enter the cell cycle when they manage to reactivate enzyme complexes that were previously stored in filaments. Indeed, after the filaments had dissolved, the cells started to divide again (*Figure 6D* and *Video 5*). Growth only stopped after several rounds of cell division, presumably because the amount of Gln1 was diluted below a critical concentration. Thus, we conclude that starvation-induced filaments can serve as storage depots for Gln1.

What is the physiological function of filament formation? To investigate this question, we monitored the regrowth of yeast after a short episode of starvation. We hypothesized that the ratio of assembled to unassembled Gln1 should specifically affect the initial growth phase of recovering yeast. To investigate this possibility, we observed the growth of Y81A mutants immediately after nutrient re-supply (the Y81A mutant displayed slower filament dissolution kinetics, but no growth defect in plating assays, see *Figure 6— figure supplements 3 and 4*). As can be seen in *Figure 6E*, the Y81A strain showed significantly slower growth after exit from starvation than wild-type cells. Thus, we conclude that exit from starvation is dependent on stored enzymes and that the rate of filament dissolution determines the timing of re-growth.

Our findings so far suggest that filament formation may regulate the ability of yeast cells to resume growth after advanced starvation conditions. But does filament formation also promote the survival of and the recovery from starvation? To investigate this possibility, we interfered with the ability of yeast to form filaments by starving them in buffers of neutral or basic pH for extended times. As can be seen in *Figure 7A*, the regrowth of yeast was severely impaired in filamentation-inhibiting buffer (pH 7 or 8), whereas cells maintained in filamentation-promoting buffer (pH 6) showed fast recovery. This suggests that filament formation by metabolic enzymes is required for recovery from severe starvation. Thus, we conclude that filament formation is a specific adaptation that enables yeast to endure severe energy depletion stress.

## Discussion

How cells survive and recover from severe starvation is a largely unresolved question. Our findings suggest that this may involve extensive changes in the organization of the cytoplasm. We demonstrate that starvation induces the self-assembly of yeast glutamine synthetase into filaments by a simple back-to-back stacking mechanism (see *Figure 7B* for a model of assembly). We further show that filament formation is triggered by starvation-induced acidification of the cytosol and results in enzyme inactivation. This catalytic inactivation is reversible, arguing that filaments are temporary storage depots for Gln1. Other filament-forming enzymes show a strikingly similar sensitivity to pH, and their growth-associated functions are probably not needed during advanced starvation. This suggests that the formation of inactive enzyme assemblies may be a general principle, which allows cells to adapt to

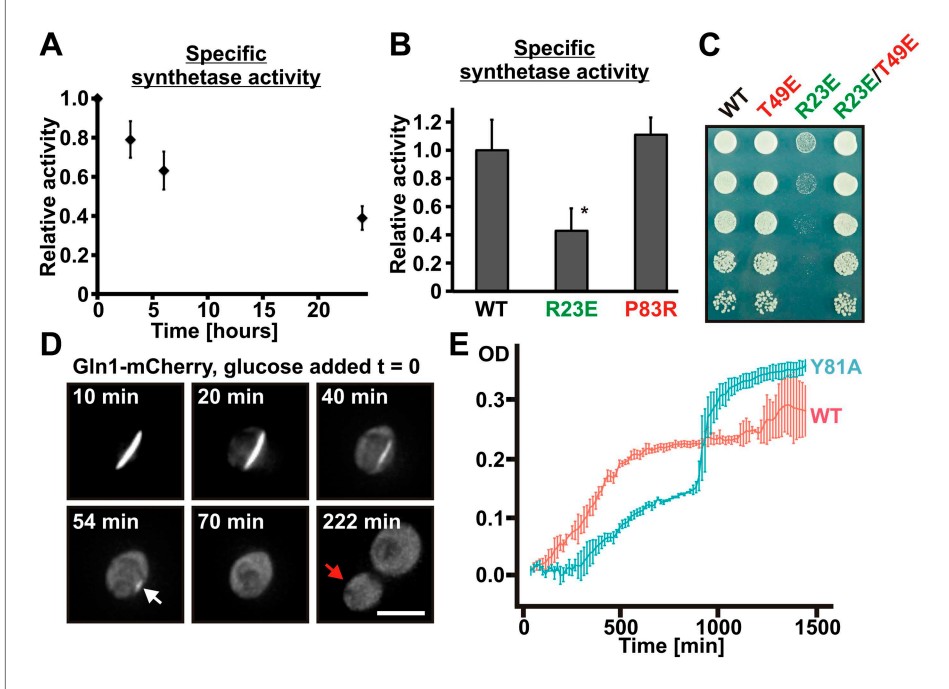

**Figure 6**. Assembled Gln1 is catalytically inactive but becomes active again after disassembly. (**A**) Lysates were prepared form wild-type cells exposed to advanced starvation conditions for 3, 6, or 24 hr, and the glutamine synthetase activity was determined as described previously (***Mitchell and Magasanik, 1984***). The obtained values were normalized to the amount of Gln1 in the cell lysate based on immunoblotting experiments. The shown data is the mean of five biological replicates (±SEM). (**B**) Lysates were prepared from yeast cells expressing wiltype or variant Gln1-V5 and the glutamine synthetase activity was determined as in (**A**). The values were normalized to the amount of Gln1 contained in the lysate based on immunoblotting experiments (*p value <0.05). (**C**) Endogenous Gln1 was substituted with the indicated wild type or variant versions expressed from an *ADH1* promoter-containing plasmid. Cells were grown over night and equal amounts of late log phase cells were spotted onto synthetic plates in serial 1:5 dilutions. (**D**) Cells expressing Gln1-mCherry from a *GAL*-inducible promoter were washed twice with water and starved in a phosphate–citrate buffer of pH 6 for 4 hr. Filament dissolution was followed after re-addition of glucose by time-lapse imaging. The red arrow denotes the newly formed bud. Note that filament dissolution and bud emergence take longer as the cells had to readjust to a new carbon source (glucose instead of galactose). The scale bar is 5 µm. Also see corresponding **Video 5**. (**E**) Wild-type and mutant (Y81A) yeast were exposed to advanced starvation conditions and regrowth was monitored after addition of nutrients (we used only 0.5% glucose in this assay as this led to slower regrowth and a more pronounced lag phase). Note that the Y81A cultures reached a slightly higher optical density, probably because of compensatory changes in the gene expression network. The values show the mean of four technical replicates. The data shown are representative of four independent experiments.

The following figure supplements are available for figure 6:

**Figure supplement 1**. Gln1 filaments form progressively after exposure of yeast to advanced starvation conditions.

**Figure supplement 2**. The R23E growth defect can be rescued by adding glutamine to the growth medium.

**Figure supplement 3**. The Y81A mutation does not affect growth.

**Figure supplement 4**. Filament dissolution is impaired in Y81A mutants.

low energy levels. Consistent with this, we found that filament formation regulates the regrowth of yeast after severe starvation. Recent findings indicate that quiescence-associated subcellular structures, such as proteasome storage granules and actin bodies, also form in a pH-dependent manner (***Peters et al., 2013***). Moreover, pH changes have been implicated in the regulation of stress-induced G protein signaling (***Isom et al., 2013***). This strongly reinforces the notion that intracellular pH changes

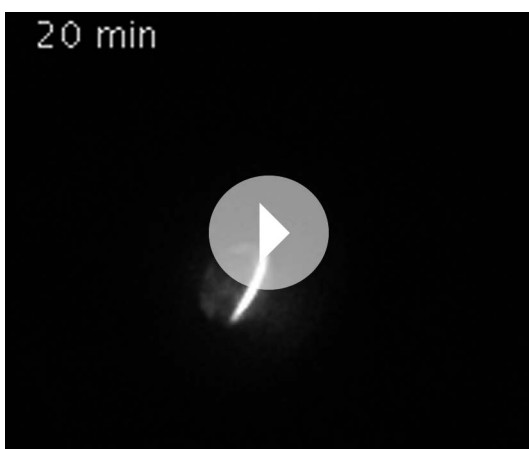

**Video 5**. Filamentous Gln1 can be reactivated upon entry into the cell cycle. Cells expressing Gln1-mCherry from a *GAL*-inducible promoter were washed twice with water and starved in a phosphate-citrate buffer of pH 6 for 4 hr. At time point 0, glucose (2%) was added and filament dissolution was followed by time-lapse microscopy. Note that filament disassembly and bud emergence take longer, because the cells had to readjust to a new carbon source (glucose instead of galactose).

serve as a global messenger to signal the depletion of energy during starvation and cellular quiescence (*Dechant et al., 2010*; *Orij et al., 2012*). In fact, pH-induced formation of filamentous structures may be a more widespread phenomenon in nature, as shown for example by the finding that spider silk formation is induced by low pH (*Kronqvist et al., 2014*).

How does the pH regulate the formation of Gln1 filaments? We noticed that Gln1 has a theoretical isoelectric point of around 6. Starvation-induced changes in the cytosolic pH will therefore strongly reduce its net charge. Thus, an overall reduction of repulsive interactions or altered charge distributions at the interface—which the mutations introduced in this study may modulate—are likely to be the driving force for assembly. An alternative scenario is that protons act as allosteric effectors that induce structural changes at the decamer–decamer interface. Accordingly, the behavior of the Gln1 mutants could also be explained by an increased or decreased propensity to undergo structural changes. However, currently we have no evidence that protonation induces large conformational changes. Regardless of the specific mechanism, what is evident is that a drop in intracellular pH induces the formation of an attractive interface at the two ends of a decameric Gln1 particle. The steric self-compatibility of this interface and the highly symmetric cylindrical structure of Gln1 then promote the assembly of Gln1 into extended filaments. Once formed, these filaments are able to associate side by side to form higher order bundles (*Figure 7B*).

Where does the energy for assembly come from? The individual stacking reactions seem to be quite stable at low pH, whereas the final filamentous structure was vulnerable to even mild perturbations. This suggests that the higher order assembly of filaments into microscopically visible fibrils is a cooperative and less favorable process. In agreement with this, we found that the formation of fibrils was strongly dependent on a macromolecular crowding agent. This indicates that the assembly reaction is not only driven enthalpically by chemical interface–interface interactions but also entropically by excluded volume effects (*Figure 7B*). Consistent with this, protein associations are often enhanced by macromolecular crowding (*Minton, 2001*), and crowding effects become particularly relevant in a self-assembling structure with multiple units. Accordingly, a recent theoretical study proposed an important role for crowding in the formation of mesoscale assemblies (*Marenduzzo et al., 2006*). This suggests that macromolecular crowding may have a key role in reorganizing the cytoplasm of starved cells.

Gln1 filaments dissolve rapidly when yeast cells re-enter the cell cycle, and the disassembled enzyme complexes are fully functional. This finding and the highly ordered structural arrangement of Gln1 into fibrils argues against the possibility that these filaments are protein aggregates consisting of misfolded proteins. Thus, we conclude that starvation-induced filament formation is not a chaotic protein aggregation event but a desired outcome of a protective cellular program. During its lifetime, a given yeast cell repeatedly transitions into and out of starvation, suggesting that on evolutionary time scales the yeast cytoplasm must have undergone numerous pH fluctuations. This makes it very likely that specific adjustments have been made to optimize the behavior of protein complexes in an energy-depleted and acidified cytoplasm. This consideration is also supported by the fact that other protein complexes assemble into distinct, starvation-induced structures that do not intermix (*Sagot et al., 2006*; *Laporte et al., 2008*; *Laporte et al., 2011*, *2013*; *Liu et al., 2012*). How this remarkable specificity of assembly is achieved in the crowded environment of quiescent cells remains to be determined, but some of the principles presented here may apply more generally.

Filament formation inactivates the enzymatic activity and results in the formation of storage depots for Gln1. How filament formation induces enzymatic inactivation is unclear, but we suspect that

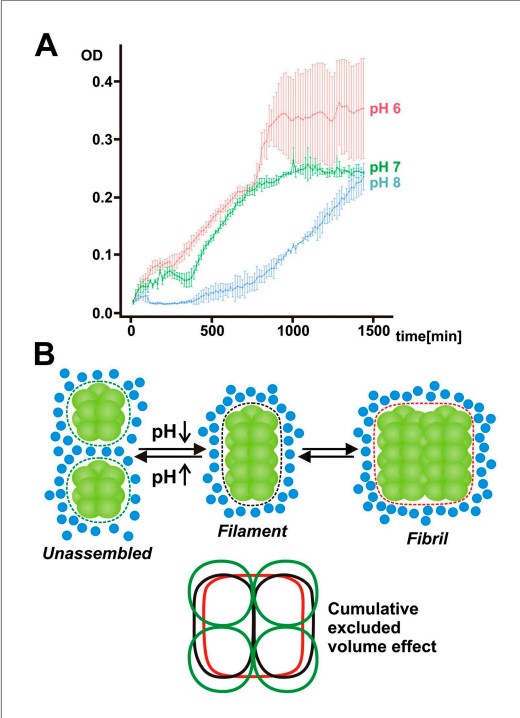

**Figure 7**. Mechanism of filament formation by Gln1 and its potential role in starvation survival. (**A**) Acidification of the yeast cytosol promotes survival and recovery from starvation. Yeast cells were exposed to advanced starvation conditions using buffers with a pH of 6, 7 or 8 for 3 days. Upon re-addition of SD medium containing a limited amount of glucose (0.5%), regrowth was monitored in a plate reader. The values show the mean of four technical replicates (±SEM). The data shown are representative of four independent experiments. (**B**) Mechanism of pH-induced filament formation by metabolic enzymes. Gln1 assembles into filaments by a back-to-back enzyme stacking mechanism. Transitions between unassembled Gln1 decamers (left), filaments (middle), and fibrils (right) are driven by pH changes and excluded volume effects. Assembly requires an acidic pH, whereas disassembly requires a basic pH. Gln1 enzyme complexes are shown in green. Blue spheres denote inert macromolecules that are excluded from the space that is occupied by Gln1. This inaccessible space is indicated by the dotted lines. The bottom diagram illustrates the cumulative excluded volume effect that entropically drives filament assembly and fibril formation. The colors of the bottom diagram correspond to the colors of the excluded volume areas above.

assembled Gln1 can no longer undergo the conformational changes that are required for enzymatic activity. It is also conceivable that the substrate cannot gain access to the catalytic site. However, when starved yeast cells are re-exposed to nutrients, intracellular pH levels rise and Gln1 filaments disassemble, thus shifting the enzyme back into its active form. In agreement with this, we found that the ability to dissolve filaments specifically affects the ability of yeast to re-grow after starvation (***Figure 6E***). Moreover, preventing the starvation-induced decline in intracellular pH severely impaired the ability of yeast to recover from prolonged starvation (***Figure 7A***), suggesting that filament formation by metabolic enzymes is required for survival of extensive energy depletion stress.

What could be the advantage of forming filaments during severe starvation? We consider four possible scenarios as likely. First, filaments may be resistant to bulk autophagy, allowing yeast cells to spare vital enzymes from degradation during prolonged starvation. Second, filament formation may promote the transition into or recovery from a more solid and probably protective material state of the cytoplasm, as has been proposed for bacteria (***Parry et al., 2014***). Third, filament formation may promote entry into a hypometabolic state that makes energy-depleted cells more resistant to metabolic fluctuations (enzyme inactivation by filament formation may provide a buffer against metabolic fluctuations, thus preventing accidental re-entry into the cell cycle). Fourth, enzyme inactivation may help conserve energy as part of a cell-wide energy conservation program, as has been proposed for mammalian cells containing ADF/cofilin filaments (***Bernstein et al., 2006***; ***Bernstein and Bamburg, 2010***). Discrimination between these possibilities has to await the results from further experimental studies.

Our findings suggest that yeast cells can extensively rearrange the cytoplasm to adjust their metabolism in response to environmental changes. We reveal the cytosolic pH as a key regulator of this process and show that assembly into higher order structures can modulate the activity of proteins. Several earlier studies reported intracellular pH changes in response to environmental perturbations. The slime mold *Dictyostelium* for example experiences a drop in cytosolic pH during starvation (***Gross et al., 1983***) and in response to stress (***Pintsch et al., 2001***). Additional findings in bacteria and metazoans point to an important role of the intracellular pH in controlling transitions into and out of hypometabolic states. Among others, pH changes have been shown to control entry into dormancy in bacteria (***Setlow and Setlow, 1980***), brine shrimp (***Busa and Crowe, 1983***; ***Hand et al., 2011***), and land snails (***Barnhart and Mcmahon, 1988***), as well as hibernation in mammalians, amphibians, and reptiles (***Malan, 2014***). The intracellular pH also regulates cellular metabolism more generally in mammalian cells (***Busa and***

*Nuccitelli, 1984*; *Moolenaar, 1986*), and promotes tumorigenesis (*Webb et al., 2011*) and the exit from quiescence (*Zetterberg and Engstrom, 1981*). These collective findings reveal intracellular pH changes as an evolutionarily ancient signal that directs developmental programs and mediates metabolic restructuring during energy limitation. It will be interesting to revisit these and other cases in the light of our findings.

## Materials and methods

### Cloning procedures
Cloning procedures were performed as described previously using the Gateway system (*Alberti et al., 2007*, *2009*). For a list of plasmids used in this study please see *Supplementary file 1A*.

### Yeast genetic techniques, strains, and media
The media used were standard synthetic media or rich media containing 2% D-glucose or 2% D-galactose. The yeast strain backgrounds were W303 *ADE+* (*leu2-3112*; *his3-11,-15*; *trp1-1*; *ura3-1*; *can1-100*; [*psi-*]; [*PIN+*]), BY4741 (*his3Δ1*; *leu2Δ0*; *met15Δ0*; *ura3Δ0*; [*psi-*]; [*PIN+*]) or a prototrophic W303 (*Klosinska et al., 2011*). The strain used in *Figure 1—figure supplements 1 and 2* is the same as used in a previous publication (*Narayanaswamy et al., 2009*). Yeast gene deletions were performed using a PCR-based approach (*Gueldener et al., 2002*). C terminal tagging of yeast genes was performed as described previously (*Sheff and Thorn, 2004*). For a list of used strains please see the *Supplementary file 1C*.

### Treatment of yeast cells
To induce filaments, yeast strains were grown in YPD until early to mid log phase. The cells were then washed once or twice with water and resuspended in a 0.1 M phosphate citrate buffer (pH 5, 6 or 7) or a 0.1 M phosphate buffer (pH 8). To ensure rapid equilibration of the intracellular and extracellular pH, some buffers contained proton carriers (75 µM monensin, 10 µM nigericin) and inhibitors to rapidly deplete cells of energy (10 mM NaN$_3$, 10 mM 2-deoxyglucose), according to what was described in a previous study (*Brett et al., 2005*). To acidify yeast cells in the presence of glucose, 2 mM 2,4-dinitrophenol was added to pH-adjusted phosphate buffers containing 2% glucose in agreement with a previously published procedure (*Dechant et al., 2010*).

### Fluorescence microscopy
General fluorescence microscopy and time-lapse videos were acquired using a Deltavision microscope system with softWoRx 4.1.2 software (Applied Precision). The system was based on an Olympus IX71 microscope, which was used with a 100x 1.4 NA (or 150x 1.45 NA) objective. The images were collected with a Cool SnapHQ camera (Photometrics) as 352x352 (or 384x384) pixel files using 1x1 (or 2x2) binning. Images were deconvolved using standard softWoRx deconvolution algorithms (enhanced ratio, high to medium noise filtering). Images were maximum intensity projections of at least 20 individual images. Representative cells are shown and each experiment was performed independently three times. Cell boundaries (indicated by the white outline in fluorescence microscopy images) were introduced by changing the contrast of the DIC image and overlaying it with the fluorescent image. Fluorophores used were mCherry or superfolder GFP with a mutation to prevent self-association (V206R).

### Treatment of yeast cells for fluorescence microscopy with FlAsH
FlAsH-labeling of tetracystein-tagged Gln1 was essentially performed as described previously (*Adams et al., 2002*; *Andresen et al., 2004*). More specifically, 200 µl of SD containing 1 µl of a 1,2-ethanediol stock (5 µM in HEPES, pH 7.5) and 4 µl of a FlAsH-EDT$_2$ stock (4 µM in 1 M Tris–HCl, pH 7.5) were inoculated with 10 µl of mid log phase yeast cells. The cells were grown overnight at 25°C. The culture was harvested by centrifugation and excess dye was washed away with PBS. The cells were incubated in PBS on a rotating wheel for 30 min and destained in enthanediol-containing PBS. Subsequently the cells were analyzed by fluorescence microscopy.

### Transmission electron microscopy (TEM) of yeast cells
Yeast cells were grown to log phase, vacuum filtered, mixed with 20% BSA, high pressure frozen (EMPACT2, Leica Microsystems, Wetzlar, Germany) and freeze-substituted with 0.1% uranyl acetate and 4% water in acetone at −90°C. Samples were transitioned into ethanol at −45°C, before infiltration into a Lowicryl HM-20 resin (Polysciences, Inc., Eppelheim, Germany), followed by UV polymerization

at −25°C. Semi-thin (150 nm thick) sections were mounted on formvar-coated mesh EM grids and stained for 3 min with lead citrate. Imaging was done in a Tecnai-12 biotwin TEM (FEI Company, Eindhoven, The Netherlands) at 100 kV with a TVIPS 2k CCD camera (TVIPS GmbH, Gauting, Germany).

## Correlative light/electron microscopy (CLEM)

For CLEM, yeast cells expressing wild type or R23E Gln1-mCherry were processed for TEM. To allow alignment of EM and light microscopy (LM) images, unstained sections on EM grids were incubated with quenched 200 nm Blue (365/415) FluoSpheres as fiducials. Grids were mounted on a glass slide with VectaShield (Vector Laboratories, Inc., Burlingame, USA) and viewed in both red (for mCherry) and UV (for the fiducials) channels. After staining for TEM, regions of interest in LM were relocated in TEM. Montaged images were acquired at multiple magnifications to facilitate the correlation. LM and TEM images were overlayed in ZIBAmira (Zuse-Institut, Berlin, Germany).

## Filament stability assay in yeast cell lysate

Gln1(R23E)-mCherry expressing yeast were grown to an $OD_{600}$ of ~0.4. The cells were harvested by centrifugation and washed twice with water. 1 ml of XL buffer (1.2 M Sorbitol, 5 mM EDTA, 0.1 M $KH_2PO_4$/$K_2HPO_4$ pH 7.5) and 12 µl 10 mg/ml lyticase were added to 10 ODs of harvested cells and the cells were incubated for 40 min at 30°C on a rotator. The spheroplasts were harvested by centrifugation and resuspended in lysis buffer (100 mM potassium phosphate, pH 6.3, 150 mM KCl, 20 mM $NaCl_2$, 5 mM $MgCl_2$, 1 mM DTT, 1% Triton X-100) containing protease inhibitors (1.25 mM benzamidin, 10 µg/ml pepstatin, 10 µg/ml chymostatin, 10 µg/ml aprotinin, 10 µg/ml leupeptin, 10 µg/ml E−64 and 0.4 mM PMSF). Cells were lysed on a bead beater for 20 min and immediately transferred onto a glass slide for analysis by fluorescence microscopy. The buffer in experiment *Figure 3B* contained 200 mg/ml Ficoll 70 and in *Figure 4A* the pH was adjusted to 5, 6, 7 or 8.

## Purification of Gln1 from bacteria and yeast

For purification of Gln1 from bacteria, wild-type and variant (E186K, R23E) Gln1 were expressed as 6xHis fusions in *E. coli* BL21 DE3. His-tagged proteins were purified to ~99% purity from bacterial lysates using a Histrap HP column (GE Healthcare) and a gradient elution profile according to the manufacturer's protocol. For purification of Gln1 from yeast, 6xHis-Gln1-V5 and 6xHis-Gln1-mCherry were expressed in W303 *ADE + gln1::SpHIS5* cells. Cell lysis was carried out in lysis buffer (50 mM $KH_2PO_4$/$K_2HPO_4$, pH 8, 150 mM KCl, 20 mM NaCl) containing protease inhibitors (1.25 mM benzamidin, 10 µg/ml pepstatin, 10 µg/ml chymostatin, 10 µg/ml aprotinin, 10 µg/ml leupeptin, 10 µg/ml E−64 and 0.4 mM PMSF). His-tagged proteins were purified at 4°C using Ni-NTA agarose according to the manufacturer's protocol for native purification in a batch format (Qiagen). Relative concentrations of purified proteins were determined by Western blotting using anti-V5 or anti-mCherry antibodies (see *Supplementary file 1B*).

## In vitro assembly of Gln1

Yeast-purified mCherry-tagged Gln1 was mixed 1:5 with a 0.1 M phosphate-citrate (pH 5 or 7) or phosphate buffer (pH 8) and incubated at room temperature for one hour. During the incubation period the samples were shaken in a thermomixer at 1000 rpm. Crowding samples contained polyethyleneglycol (average molecular weight 20 kDa) at a concentration of 70 mg/ml. After one hour the samples were transferred onto a glass slide and analyzed by fluorescence microscopy.

## Dynamic light scattering (DLS) of purified protein

DLS measurements were performed using a Zetasizer instrument (Malvern). 100 µg of wild-type or variant Gln1 were diluted in equal volumes of a HEPES buffer (pH 6.5) containing 150 mM KCl, 20 mM NaCl, 5 mM MgCl and 1 mM DTT in a low binding microcentrifuge tube. The samples were rotated for 1 hr at room temperature. 50 µl of the samples were transferred into a small volume DLS cuvette and placed inside the measurement chamber. The shown values are average values obtained from 20 measurements at room temperature.

## Gel filtration of purified proteins

100 µg of purified wild-type and variant Gln1 was loaded onto a Superose 6HR 10/30 column (GE Healthcare). Eluates were collected as 300 µl fractions and were applied onto a nitrocellulose membrane by using the mini-fold dot blot system (Whatman). Proteins on the membrane were detected by immunoblotting with an anti-His antibody (see *Supplementary file 1B*).

### Thioflavin T staining of yeast cells

ThT staining of yeast cells containing Gln1 filaments was essentially performed as described previously (*Alberti et al., 2010*).

### Semi-denaturing detergent agarose gel electrophoresis

Semi-denaturing detergent agarose gel electrophoresis (SDD-AGE) with lysates from filament-containing cells was performed essentially as described previously (*Alberti et al., 2010*).

### Far-uv circular dichroism

Samples were measured in a 1 mm cuvette on a JASCO J-815 spectropolarimeter. Samples were diluted 1:3 with a phosphate–citrate buffer of pH 7.4 or 6 and scaled to the same concentration as determined from UV-Vis absorption at 280 nm.

### Negative staining of purified 6xHis-tagged Gln1(R23E) for TEM

Different dilutions of purified 6xHis-tagged Gln1(R23E) were prepared in buffer A (100 mM potassium phosphate, pH 6.3, 150 mM KCl, 20 mM NaCl, 5 mM $MgCl_2$, 1 mM DTT). 3 µl of the sample was deposited for 1 min on glow-discharged, carbon-coated EM mesh grids. After blotting, samples were fixed with 1% glutaraldehyde in water. The grids were washed once in water and stained with 2% uranyl acetate. After drying, grids were imaged in TEM (same setup as used for TEM on yeast cells).

### Glutamine synthetase activity assay

Synthetase assays were carried out with yeast cell lysate as described previously (*Mitchell and Magasanik, 1984*).

## Acknowledgements

We thank Tony Hyman, Titus Franzmann and several other members of the MPI-CBG for critical reading of the manuscript. We are grateful to the MPG and the DIGS-BB program for funding.

## Additional information

### Funding

| Funder | Grant reference number | Author |
|---|---|---|
| Max Planck Society | | Ivana Petrovska, Elisabeth Nüske, Gayathrie Kulasegaran, Liliana Malinovska, Sonja Kroschwald, Doris Richter, Kimberley Gibson, Jean-Marc Verbavatz, Simon Alberti |
| DIGS-BB | | Matthias C Munder |
| Helmholtz Association | | Karim Fahmy |

The funders had no role in study design, data collection and interpretation, or the decision to submit the work for publication.

### Author contributions

IP, EN, MCM, LM, J-MV, Conception and design, Acquisition of data, Analysis and interpretation of data; GK, SK, DR, KF, KG, Acquisition of data, Analysis and interpretation of data; SA, Conception and design, Acquisition of data, Analysis and interpretation of data, Drafting or revising the article

## Additional files

### Supplementary files

• Supplementary file 1. (A) Plasmids used in this study. (B) Antibodies used in this study. (C) Yeast strains used in this study.

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
