## [Decision Letter]

Thank you for sending your work entitled “Filament formation by metabolic enzymes is a specific adaptation to an advanced state of cellular starvation” for consideration at *eLife*. Your article has been favorably evaluated by a Senior editor and 3 reviewers, one of whom is a member of our Board of Reviewing Editors.

The following individuals responsible for the peer review of your submission have agreed to reveal their identity: Jeffery W Kelly, Reviewing editor; Daniel Otzen, peer reviewer.

The Reviewing editor and the other reviewers discussed their comments before we reached this decision, and the Reviewing editor has assembled the following comments to help you prepare a revised submission.

The authors present a very nice study of the yeast metabolic protein Gln1, which forms filaments under starvation conditions. They convincingly demonstrate that Gln1 forms intracellular filaments in response to starvation-induced cytosolic acidification. They go on to suggest that Gln1 filaments form by back-to-back stacking of native or near-native structures and show that filament formation is important for successful recovery from starvation. We will carefully consider a revised paper addressing rudimentary biophysical and structural insights into filament formation, as specifically suggested by the reviewers below.

1) The structural basis for self-assembly and enzyme inactivation is lacking in the current manuscript. What could be done quite easily by the authors to partly remedy this deficiency is to show that the near-native homo-decameric structure of Gln1 is largely retained in the filament by Far-UV circular dichroism or FT-IR or an analogous structural fingerprinting method, where the non-filamentous and filament structures could be compared.

2) It would also be useful to rule out cross-beta-sheet assembly for the basis of inactivation-as forms in the native amyloid structures (Curli, pMel17, etc.)

3) Could the authors show how purified enzyme filaments upon dropping the pH (e.g., measured through DLS or simple turbidity) and how quickly this is reversed by restoring pH? This would provide even more of a link between molecular mechanisms and cell physiology.

4) The authors should report the standard deviation of their measurements of time of filament formation and dissolution in addition to the mean.

5) The authors convincingly demonstrate that R23E forms filaments in cells with or without a tag by CLEM/EM. Did they try wild type? It would be pleasing to see the absence of such structures in nonstarved cells expressing wild type and the appearance of such structures upon starvation.

6) The authors show very convincingly that neutral media prevents the formation of filaments even under starvation conditions. Do natural yeast typically grow in acidic conditions? If not, what is the relevance of filament formation to yeast life? Do Gln1 sequences from other yeasts (or sequenced wild *S. cerevisiae* isolates) contain mutations that abolish the filament-forming phenomenon?

7) The authors suggest that a pH-induced conformational change is the most likely mechanism of pH-dependent filament formation. Their second scenario, in which abrogation of Gln1 charge enables association is much more likely given the data (isoelectric point near 6 and charge-altering mutations that influence filament formation).

8) The authors state that “Gln1 filaments are rather unstable...” and go on to argue that the assembly reaction must be highly cooperative. I don't think that the thermodynamic stability of the filaments was measured at any pH. The fact that they disassemble at higher pHes doesn't mean anything regarding their stability at low pH. It's likely that the filaments are highly stable at low pH (so assembly is very favorable) and unstable at high pH (so disassembly is favorable). The authors should revisit this paragraph.

---

## [Author Response]

*1) The structural basis for self-assembly and enzyme inactivation is lacking in the current manuscript. What could be done quite easily by the authors to partly remedy this deficiency is to show that the near-native homo-decameric structure of Gln1 is largely retained in the filament by Far-UV circular dichroism or FT-IR or an analogous structural fingerprinting method, where the non-filamentous and filament structures could be compared*.

We agree and have performed Far-UV CD experiments as suggested. As now shown in Figure 4—figure supplement 2, exposure to an acidic buffer does not induce extensive structural changes in Gln1, despite the fact that the same treatment leads to assembly (see Figure 4—figure supplement 1). This is consistent with the interpretation that Gln1 decamers largely remain intact in the assembled state. We also performed electron microscopy on negatively stained R23E Gln1 isolated from yeast cells. In these samples, we identified filaments with a diameter of ∼120 Å (Figure 2—figure supplement 6). These filaments are formed by a structural unit that roughly repeats every 100 Å. These size proportions are consistent with the previously reported dimensions of Gln1 decamers (22). This indicates that the structure of Gln1 remains intact and that assembly proceeds by the proposed back-to-back stacking mechanism.

*2) It would also be useful to rule out cross-beta-sheet assembly for the basis of inactivation-as forms in the native amyloid*
*structures (Curli, pMel17, etc.)*

This is a good point. We have performed two different experiments to demonstrate that filament formation does not involve the formation of cross-β structure. First, we subjected lysates from yeast to semi-denaturing detergent- agarose gel electrophoresis (SDD-AGE). This method identifies cross-β structures based on their resistance to detergents. As can now be seen in Figure 2—figure supplement 5, filament formation is not associated with an increased resistance to detergent. We further treated yeast cells containing Gln1 filaments with Thioflavin T (ThT), a dye that specifically binds to cross-β structure. As shown in Figure 2—figure supplement 4, the Gln1 filaments could not be stained with ThT, whereas control cross-β filaments showed strong ThT binding (these control samples were processed in parallel in the same experiment). Based on these findings, we conclude that filamentation does not involve the formation of cross-β structure.

*3) Could the authors show how purified enzyme filaments upon dropping the pH (e.g., measured through DLS or simple turbidity) and how quickly this is reversed by restoring pH? This would provide even more of a link between molecular mechanisms and cell physiology*.

As suggested by the reviewers, we followed the assembly of Gln1 over time by dynamic light scattering. We found that wildtype Gln1 gradually transitioned into a high molecular weight form in acidic buffer, whereas mutant Gln1 (E186K) was unaffected by the same treatment (Figure 4—figure supplement 1). We also performed experiments to reverse Gln1 assembly by raising the pH, but we found that disassembly was incomplete. However, we do not interpret this as evidence that the *in vitro* structures are different from the ones formed *in vivo*, mainly for two reasons. First, mutations that abrogated filament formation *in vivo* also prevented filament formation *in vitro* (see Figures 3 and 4), strongly suggesting that the *in vitro* structures are physiological. Second, pH experiments to dissolve preformed filaments in permeabilized yeasts were also unsuccessful, suggesting that a pH increase alone is not sufficient to fully reverse filamentation. So far complete reversal could only be achieved by exposing previously starved cells to glucose. Therefore, it seems that in addition to a change in pH, complete reversal requires other factors, such as ATP, small metabolites or the action of molecular chaperones. Experiments to investigate the mechanism of filament disassembly are underway and will be reported elsewhere.

*4) The authors should report the standard deviation of their measurements of time of filament formation and dissolution in addition to the mean*.

We now mention the standard deviation in the text.

*5) The authors convincingly demonstrate that R23E forms filaments in cells with or without a tag by CLEM/EM. Did they try wild type? It would be pleasing to see the absence of such structures in nonstarved cells expressing wild type and the appearance of such structures upon starvation*.

The CLEM experiments with untagged R23E were successful for two reasons: (1) R23E Gln1 was overexpressed from a plasmid, and (2) we performed these experiments with dividing cells, in which the cytoplasm is relatively disordered. To be able to identify untagged wildtype Gln1, we had to perform experiments with starved cells to induce filament formation. We noticed, however, that the cytoplasm of starved yeast contained many structures that were not present in actively dividing cells. Given the multiplicity of ultrastructural changes, we were unable to reliably identify the specific structures that are caused by Gln1.

*6) The authors show very convincingly that neutral media prevents the formation of filaments even under starvation conditions. Do natural yeast typically grow in acidic conditions? If not, what is the relevance of filament formation to yeast life? Do Gln1 sequences from other yeasts (or sequenced wild* S. cerevisiae *isolates) contain mutations that abolish the filament-forming phenomenon?*

Yes, budding yeast generally prefer acidic growth media. Growth is aided by a low pH, and the growth rate slows down when the pH shifts to the neutral or basic range. Moreover, yeasts acidify their media as a result of metabolic activity. Thus yeast cells prefer and actively create an acidic environment that is favorable for growth. As a consequence, however, growing yeasts have to expend energy to remove protons from the cytosol. Therefore, even small fluctuations in energy status can lead to fluctuations in intracellular pH. Yeast may, however, be special in this regard, because other organisms have found ways to acidify their cytoplasm independently of the outside pH (21).

We have performed a multiple sequence alignment of *Saccharomyces cerevisiae* Gln1 with other fungal species. The critical residues required for filament formation are largely conserved in the group of *Saccharomyces*, *Candida* and *Kluyveromyces*. Identical mutations as introduced in this study were identified in distantly related fungi, for example in *Schizosaccharomyces japonicus* (P83R), *Aspergillus nidulans* (E186K) and *Neurospora crassa* (E186K). However, these fungi also carry mutations in proximity to the newly introduced amino acids, so it is possible that these changes have been compensated for by mutations elsewhere in the protein. Based on these findings we cautiously conclude that the filament-forming abilities of Gln1 are conserved and not selected against.

*7) The authors suggest that a pH-induced conformational change is the most likely mechanism of pH-dependent filament formation. Their second scenario, in which abrogation of Gln1 charge enables association is much more likely given the data (isoelectric point near 6 and charge-altering mutations that influence filament formation)*.

This has been addressed by making changes to the text.

*8) The authors state that “Gln1 filaments are rather unstable...” and go on to argue that the assembly reaction must be highly cooperative. I don't think that the thermodynamic stability of the filaments was measured at any pH. The fact that they disassemble at higher pHes doesn't mean anything regarding their stability at low pH. It's likely that the filaments are highly stable at low pH (so assembly is very favorable) and unstable at high pH (so disassembly is favorable). The authors should revisit this paragraph*.

Our comment was not so much about the influence of the pH on filament stability but about the fact that the integrity of the filaments was dependent on macromolecular crowding. So while we agree with the reviewer that the pH-induced back-to-back stacking interactions may actually be quite stable, additional interactions – in particular those between individual filaments – may be rather weak. We have revamped the paragraph to make it more obvious that this statement refers to the overall morphology of the filaments and not to individual stacking reactions.